# The Role of TRAIL in Apoptosis and Immunosurveillance in Cancer

**DOI:** 10.3390/cancers15102752

**Published:** 2023-05-13

**Authors:** Julio M. Pimentel, Jun-Ying Zhou, Gen Sheng Wu

**Affiliations:** 1Molecular Therapeutics Program, Karmanos Cancer Institute, School of Medicine, Wayne State University, Detroit, MI 48201, USA; pimentej@karmanos.org (J.M.P.); zhouj@karmanos.org (J.-Y.Z.); 2Cancer Biology Program, School of Medicine, Wayne State University, Detroit, MI 48201, USA; 3Department of Oncology, School of Medicine, Wayne State University, Detroit, MI 48201, USA; 4Department of Pathology, School of Medicine, Wayne State University, Detroit, MI 48201, USA

**Keywords:** TRAIL, apoptosis, resistance, immunosurveillance, PD-L1

## Abstract

**Simple Summary:**

Tumor necrosis factor (TNF)-related apoptosis-inducing ligand (TRAIL) plays an important role in apoptosis and tumor immunosurveillance. Because TRAIL selectively induces apoptosis in tumor cells, there is growing interest in using it as a cancer therapy agent, but the development of TRAIL resistance has limited its clinical development. Recent evidence suggests that the TRAIL pathway can activate the immunological checkpoint protein programmed death-ligand 1 (PD-L1), which has recently been found to play an important role in TRAIL resistance and tumor invasion. Thus, targeting PD-L1 could be a promising new therapeutic strategy to improve TRAIL-based treatments in human cancers.

**Abstract:**

Tumor necrosis factor (TNF)-related apoptosis-inducing ligand (TRAIL) is a member of the TNF superfamily that selectively induces apoptosis in tumor cells without harming normal cells, making it an attractive agent for cancer therapy. TRAIL induces apoptosis by binding to and activating its death receptors DR4 and DR5. Several TRAIL-based treatments have been developed, including recombinant forms of TRAIL and its death receptor agonist antibodies, but the efficacy of TRAIL-based therapies in clinical trials is modest. In addition to inducing cancer cell apoptosis, TRAIL is expressed in immune cells and plays a critical role in tumor surveillance. Emerging evidence indicates that the TRAIL pathway may interact with immune checkpoint proteins, including programmed death-ligand 1 (PD-L1), to modulate PD-L1-based tumor immunotherapies. Therefore, understanding the interaction between TRAIL and the immune checkpoint PD-L1 will lead to the development of new strategies to improve TRAIL- and PD-L1-based therapies. This review discusses recent findings on TRAIL-based therapy, resistance, and its involvement in tumor immunosurveillance.

## 1. Introduction

Tumor necrosis factor (TNF)-related apoptosis-inducing ligand (TRAIL) is a cytokine of the TNF superfamily that plays an important role in apoptosis and tumor surveillance. TRAIL was discovered by comparing the sequence of its c-terminal (CTE) domain to that of other TNF superfamily members, such as FasL and TNF-α [1,2]. Since then, TRAIL has been shown to be expressed by immune cells as a homotrimeric type 2 transmembrane or soluble protein. TRAIL binds to its two death receptors, DR4 and DR5, to induce apoptosis in tumor cells without harming normal cells [3,4]. Because of these features, the TRAIL pathway is regarded as a more appealing cancer therapy agent than FasL and TNF-α, both of which activate similar pathways but cause unacceptable systemic toxicity when administered. The safety of TRAIL-based therapy in cancer patients has been well established in Phase I and Phase II clinical trials [5,6,7]. However, the development of resistance to TRAIL-based therapies and poor pharmacokinetic profiles limit their clinical utility [8]. Therefore, novel approaches are needed to improve the pharmacokinetic profile of TRAIL-based therapies and reduce resistance to TRAIL-based treatment. 

## 2. TRAIL Ligand

TRAIL is a type 2 transmembrane protein with two forms: membrane-bound and solubilized [9]. Membrane-bound TRAIL is a native form of TRAIL that consists of the C-terminal extracellular domain (CTE), the transmembrane (TM), an extracellular stalk (ES) region, and N-terminal cytoplasmic domain (NTC) [9]. The CTE domain is conserved and homologous to CD95L and TNF-α. When the ES region (aa 89–106) is catalyzed by cathepsin E, TRAIL detaches from the cell membrane [10]. Consequently, the TRAIL monomer is assembled and converted into a soluble homotrimer composed of monomers arranged in a jellyroll topology with two antiparallel beta sheets connected by a zinc atom [11,12,13]. The zinc atom chelates the cysteine (cys)-230 sulfhydryl group in each TRAIL monomer, linking them in the trimeric core [13]. Therefore, cys-230 plays an important role in the trimeric structure of TRAIL while distinguishing it from other members of the TNF family [13,14]. Many tissues express TRAIL, with the immune and lymphatic systems having high levels [15], suggesting the role of TRAIL in regulating immune surveillance. 

## 3. TRAIL Receptors

There are five receptors for TRAIL, including membrane-bound receptors DR4 (TRAIL-R1, TNFRSF10A, CD261, APO2), DR5 (TRAIL-R2, TNFRSF10B, CD262, KILLER, TRICK2, ZTNFR9), DcR1(TRAIL-R3, TRID), DcR2 (TRAIL-R4, TRUNDD), and the soluble receptor osteoprotegerin (OPG) [16,17,18,19,20,21]. DR4 and DR5 are death receptors. TRAIL binds to and activates DR4 and DR5 to induce apoptosis. DR4 and DR5 are type 1 transmembrane proteins that comprise a cysteine-rich extracellular domain, a transmembrane domain, and an intracellular domain that contains a death domain (DD) [22,23]. The first cys of the extracellular domain contains a preligand assembly domain (PLAD), which promotes the oligomerization of the DR4 and DR5 trimers, and thus, improves TRAIL-DR4/5 binding [24]. O-glycosylation and other post-translational modifications of DR4 and DR5 have also been shown to enhance the binding of TRAIL-DR4/DR5 by stabilizing the assembly of death receptors and preventing endocytosis [25]. Although the regulation of death receptor expression is not fully understood, the tumor suppressor p53 has been shown to regulate the expression of the TRAIL death receptor [19,26,27]. While an initial study showed that the chemotherapeutic agent doxorubicin and etoposide induce DR5 [19], subsequent studies have shown that several agents can induce the expression of DR4 and DR5, which can sensitize cancer cells to chemotherapy and radiation therapy [6]. 

In addition to the TRAIL death receptors DR4 and DR5, there are three TRAIL decoy receptors, DcR1, DcR2, and OPG. DcR1 and DcR2 are cell surface receptors; the former is a glycosylphosphatidylinositol (GPI) anchored receptor lacking DD and the latter has a functionally inactive truncated DD [6,9,28]. In contrast, OPG is a secreted receptor with DD that inhibits apoptosis [9]. Thus, these decoy receptors can compete for TRAIL binding to DR4 and DR5, thus inhibiting TRAIL-induced apoptosis [22]. 

## 4. The TRAIL Apoptosis Pathway 

The first step in the TRAIL apoptosis pathway is the formation of death-inducing signaling complexes (DISCs), which are initiated by recruiting Fas-associated death domain (FADD) adapter proteins. When TRAIL binds to DR4 and DR5, the conformation of DR4 and DR5 changes, which promotes interaction through the death domains between DR4 and DR5 with FADD (Figure 1). Once FADD binds to DR4 and DR5, it readily recruits cysteine protease precursors known as procaspases 8/10 [29,30]. Procaspases 8/10 are known to have an N-terminal pro-domain, two death effector domains (DED1/DED2), and a C-terminal protease domain with large (p20/p18) and small (p12/p10) subunits linked by a short linker region [29,30]. Previous studies on the protein sequence alignment of the two caspases revealed that caspase 10 differs from caspase 8 in cleavage sites in the short linker region and subunit size [31]. It is unknown whether two death receptors play an equal role in inducing apoptosis. Studies suggest that DR5 binds to TRAIL more efficiently than DR4 through a stepwise binding mechanism [32,33]. Although leukemic cells prefer to initiate apoptosis through DR4 [34,35], many cancer cells, including those from colorectal cancer, are equally sensitive to DR4- and DR5-induced apoptosis [36,37]. Therefore, more research is needed to better understand the conditions under which one or both death receptors are preferred over another in inducing apoptosis in cancer cells.

When bound to FADD, DED1s dimerize procaspases 8/10, resulting in autocatalytic cleavage and activation of caspases 8/10 [38]. Caspases 8/10 are regulated by a cellular FLICE-inhibitory protein (c-FLIP), a cellular FADD-like interleukin (IL)-1-converting enzyme (FLICE)-inhibitory protein [39]. c-FLIP is a DED-containing protein that is structurally similar to caspases 8/10 but lacks protease activity due to the absence of a critical NH2 amino acid residue at the active site [39]. c-FLIP inhibits caspase activation by interfering with the interaction between FADD and procaspases 8/10. Active caspases 8/10 can directly activate caspases 3, 6, and 7 to induce cell death.

Furthermore, activated caspases 8/10 can enhance apoptosis by cleaving the B-cell lymphoma-2 (BCL-2) family protein BID [40,41]. BID is a cytosolic protein that is cleaved by activated caspase 8 on the Asp-60 residue into two fragments: c-terminal (truncated BID [tBID], p15) and p7 [40,41]. tBID activates the pro-apoptotic proteins BAX and BAK [42]. Active BAX/BAK undergoes a conformational change, resulting in dimerization and the formation of pores in the mitochondrial outer membrane (MOM) or MOM permeabilization (MOMP) [43]. MOMP induced by BAX/BAK releases cytochrome c (Cyt c) and other factors, including the DIABLO homolog (second mitochondrial-derived activator of caspases [SMAC/DIABLO]) from the mitochondria [44] to the cytosol, where it combines with ATP and the adaptor protein apoptosis-protease activating factor 1 (Apaf-1) to form an apoptosome [45,46]. As a result, caspase 9 is activated [46,47]. Furthermore, SMAC/DIABLO can increase apoptosome formation by inhibiting anti-apoptotic proteins called inhibitors of apoptosis (IAP) [44]. Active caspase 9 activates caspases 3, 6, and 7 in the same way that caspases 8/10 do [48].

Caspase 3 activation affects several downstream substrates, resulting in DNA fragmentation and cell disintegration. An example is caspase-activated DNase (CAD)/DNA fragmentation factor [DFF]). CAD is a caspase-3-activated endonuclease activated by the proteolytic cleavage of the CAD inhibitor (ICAD, DFF45) [49,50]. Activated CAD can then dimerize and bind to A/T-rich DNA regions, resulting in double-stranded DNA fragments [50,51]. The deactivation of DNA repair proteins, such as poly (ADP-ribose) polymerase-1 (PARP-1) can also increase CAD-dependent DNA fragmentation [52,53]. Furthermore, caspase 3 can deactivate several other survival proteins to enhance apoptosis, including those involved in nuclear structure maintenance, transcription, cell cycle, membrane-bound, cell adhesion, cell–cell communication, RNA synthesis/splicing, and protein translation/modification [53]. As a result, cells eventually disintegrate and form apoptotic bodies, which are consumed by phagocytic cells [54]. 

## 5. TRAIL-Mediated Non-Apoptotic Signaling

In addition to the induction of apoptosis, TRAIL has also been shown to activate several non-apoptotic signaling pathways. Among these pathways are the extracellular signal-regulated kinase (ERK) [55,56,57], AKT [56,58,59] and NF-κB [60,61,62] pathways. These pathways are usually cell-type specific, and the activation of these pathways inhibits apoptosis [55,57,58,61,63,64]. For example, a previous study suggests that TRAIL can activate the ERK, AKT, and NF-κB pathways through a secondary complex that forms after the formation of DISC [65]. This complex contains FADD, caspase-8, c-FLIP, receptor-interacting protein 1 (RIP1), TNF receptor-associated factor 2 (TRAF2), IκB kinase, TNF receptor 1-associated death domain protein (TRADD), and NF-κB essential modulator (NEMO) [60,65,66,67]. Furthermore, the location of death receptors affects whether apoptosis occurs. It has been demonstrated that death receptor distribution in lipid rafts induces apoptosis, while non-apoptotic signaling can occur outside these rafts [68,69]. However, the precise mechanisms and circumstances that drive TRAIL to promote the formation of secondary complexes are still being explored. Furthermore, TRAIL can generate a tumor-supportive immune microenvironment by producing cytokines/chemokines, including CXCL1, CXCL5, CCL2, IL-8, and NAMPT, to polarize monocytes to M2-like cells [70]. Furthermore, TRAIL death receptor 2 is overexpressed in KRAS-mutated tumors, and this overexpression activates the Rac1/PI3K pathway, which in turn promotes KRAS-driven tumor progression, invasion, and metastasis [71]. A recent study showed that TRAIL could induce the expression of the immune checkpoint protein programmed death-ligand 1 (PD-L1) via the ERK pathway, and that inhibiting it made TRAIL-resistant cells susceptible to TRAIL-induced apoptosis [57]. These findings show that the TRAIL pathway can activate oncogenic signaling pathways and immunological checkpoint responses and produce cytokines/chemokines that promote cancer cell survival.

## 6. TRAIL Resistance Mechanisms

The clinical development of TRAIL-based cancer therapy faces several challenges. To begin with, many cancer cells are intrinsically resistant to TRAIL. Second, previously sensitive cancer cells develop resistance to TRAIL (acquired resistance). Third, no patient population that can benefit from TRAIL has been selected. Finally, the mechanisms underlying the resistance of TRAIL are not fully understood [6]. Therefore, these features warrant further investigation into the mechanisms that confer TRAIL resistance to develop TRAIL-based therapies for clinical use.

Increasing evidence suggests that TRAIL resistance mechanisms are diverse and can occur anywhere along the TRAIL signaling pathway, from the cell surface to downstream caspases (Figure 2). Specifically, resistance to TRAIL can be conferred by the dysfunction, degradation, or polymorphisms of the death receptor on the cell surface, resulting in reduced binding of TRAIL to its death receptors and increased cancer cell survival. For example, a loss-of-function mutation of DR5, found in human head and neck cancer, is TRAIL resistant [72]. Furthermore, previous studies using TRAIL death receptor knockout mice with diethyl nitrosamine-induced liver tumors or lymphoma showed increased cancer metastasis [73]. Based on these findings, several therapeutics, including the use of proteasome (to prevent death receptor degradation) and histone deacetylase (HDAC) (to block death receptor acetylation) inhibitors, have been proposed [74,75]. 

DcR (DcR1, DcR2, and OPG) expression can also confer TRAIL resistance by competing for the binding of TRAIL to DR4 and DR5 to inhibit apoptosis [76,77,78]. It has been shown that TRAIL-DcR2 binding activates tumor-promoting downstream pathways in cervical cancer cells and the NF-κB pathway in large granular lymphocyte leukemia [79,80]. DcR1 has also been found in lipid rafts and has been shown to inhibit the formation of DISC associated with DR5-TRAIL [76]. Therefore, these findings suggest that targeting TRAIL receptors has an important implication in the prevention of cancer and the induction of apoptosis. 

Furthermore, when TRAIL-DR4/DR5 is activated, proteins such as c-FLIP are recruited to DISC and replace procaspases 8/10, forming an inactive complex [81]. Because of this, caspases 8/10 are rendered inactive. In this regard, the c-FLIP protein has been found to be overexpressed in human cancers, including prostate cancer (DU145) and non-small cell lung cancer (A549) cells, and has been linked to poor prognoses [82]. In addition, caspases 8/10 mutations can confer TRAIL resistance in cancer cells. A p10 mutation in procaspase 8 was found in acute myeloid leukemia, impairing the dimerization of procaspase 8 [83]. Caspase 10 mutations have also been found in colon, gastric, and NHL cancers, leading to TRAIL-induced apoptosis resistance [84]. Finally, though rare, mutations in the DISC-forming FADD protein can confer TRAIL resistance in cancer cells such as NSCLC cells [85]. Thus, several mutations in the DISC protein can confer resistance to TRAIL by preventing the activation of downstream mechanisms that lead to extrinsic and intrinsic apoptosis pathways.

In addition, anti-apoptotic proteins can promote TRAIL resistance. BCL-2 and BCL-XL proteins, for instance, inhibit TRAIL-induced apoptosis [86,87]. Several studies have shown that small-molecule BCL-2 inhibitors can be used to inhibit these proteins [88]. Members of the IAP family, including XIAP (X-linked inhibitor of apoptosis protein) and survivin, which inhibit caspase 9 and caspase 3 activity, are negative regulators of the TRAIL apoptosis pathway. Targeting IAP expression has been shown to sensitize cancer cells to TRAIL-induced cell death [89,90], suggesting that in cancer cells whose IAP is overexpressed, inhibition of IAP is a strategy to overcome resistance to TRAIL. Although several resistance mechanisms to TRAIL have been identified [91], a complete understanding of these resistance mechanisms is still needed to develop better cancer therapies.

## 7. Targeting TRAIL and TRAIL Death Receptors for Cancer Therapies

Because TRAIL selectively induces apoptosis in cancer cells, clinical trials were conducted to test the efficacy of TRAIL and agonist antibodies targeting death receptors in cancer patients (Table 1) [7,92]. Unlike most chemotherapeutic drugs, TRAIL ligand-based therapy causes apoptosis in tumor cells in a p53-independent manner [22]. Therefore, several TRAIL-based cancer monotherapies and combinations have been tested in human clinical trials [7,92]. The earlier forms of TRAIL used were recombinant TRAIL (rTRAIL), which was purified with a poly-Histidine (His) or FLAG epitope (FLAG-TRAIL) tag in the NTC domain [93]. Despite promising in vitro/vivo results, His-and FLAG-tagged rTRAIL aggregated and caused hepatotoxicity in vitro [3,93,94,95]. An example of rTRAIL used in clinical trials is Dulanermin (Apo2L.O or AMG-915) [93]. Dulanermin can form stable bioactive trimers that bind to DR4 and DR5 and induce apoptosis. Dulanermin has previously been shown to selectively induce apoptosis in cancer cells and work in tandem with a number of chemotherapeutic agents [3,96]. Dulanermin has been evaluated in non-small cell lung cancer (NSCLC) (phase III: NCT03083743) and B-cell non-lymphoma Hodgkin’s (phase II: NCT01258608) [7,97]. The addition of Dulanermin (i.v., 75 g/kg) to standard therapy (cisplatin [i.v., 30 mg/m^2^] or Vinorelbine [i.v., 25 mg/m^2^]) improved progression-free survival (PFS) (6.4 months) in patients with advanced untreated NSCLC compared to placebo with standard therapy (3.5 months) [97]. Furthermore, the combination demonstrated acceptable toxicity while increasing the overall response rate (ORR) in the Dulanermin arm [97]. Despite promising results, Dulanermin has limitations, including a poor pharmacokinetic (PK) profile, a short half-life, and the ability to bind to DcRs [98,99].

The limitations of rTRAIL have led to research on novel ways of improving its therapeutic efficacy. The conjugation of rTRAIL with cytotoxic agents is one strategy for enhancing rTRAIL’s therapeutic efficacy. rTRAIL combined with chemotherapy has been shown to improve clinical outcomes in Phase III clinical studies for patients with advanced NSCLC [7,97]. Furthermore, TRAIL-based therapy can be combined with immunotherapy (NCT02991196), biological treatment (NCT00400764, NCT00508625), and targeted therapy (NCT01258608, NCT00315757) to improve overall survival (OS) in a variety of cancers [7,92]. 

Despite new drug combinations and formulations, rTRAIL’s poor pharmacokinetic profile continues to limit its clinical development, including its short biological half-life [100,101]. Therefore, efforts have been made to overcome these limitations to improve rTRAIL’s poor bioactivity, stability, and tumor specificity. For example, the addition of a tenascin C oligomerization domain, the formation of single-chain TRAIL trimers, and the covalent attachment of TRAIL to molecules (e.g., human serum albumin [HSA] and polyethylene glycol) have been used to improve the stability of rTRAIL [93]. An example of trimeric forms of the TRAIL protein is SCB-313, which was tested in Phase I clinical trials for peritoneal malignancies (NCT03443674 and NCT04051112) [102]. Furthermore, ABBV-621 (Eftozanermin) is a hexavalent TRAIL-Fc fusion protein that has shown antitumor activity with acceptable toxicity in Phase I clinical trials in solid tumors (NCT03082209) [103]. Currently, ABBV-621 is being studied in Phase II clinical trials for multiple myeloma (NCT04570631). One challenge with TRAIL-based therapy is its distribution to tumors. To address this, the combination of TRAIL with nanoparticles is a unique approach to improving the delivery of TRAIL to tumors [104]. Several TRAIL-containing nanoparticles have been developed, with compositions ranging from human serum albumin [104,105,106] to poly (lacto-co-glycolic) acid (PLGA) microspheres [107,108] to liposomes [109,110]. Thus, more research is needed to fully comprehend the clinical potential of nanoparticles containing TRAIL. Despite these modified forms of rTRAIL showing promising in vitro/in vivo results, such as a better pharmacokinetic profile, longer half-life, and higher clearance rate, the delivery of rTRAIL to tumors has remained challenging [93].

In contrast, TRAIL agonist antibodies against its death receptors have several advantages over TRAIL ligands, including directly targeting DR4 or DR5 and being more stable [111]. Therefore, the focus of studies has shifted to the development of agonistic antibodies against death receptors for treating cancer. Antibodies against death receptors are classified into two groups: (1) DR4-targeting antibodies (e.g., mapatumumab) [112]; and (2) DR5-targeting antibodies (e.g., conatumumab, lexatumumab, tigatuzumab, drozitumab, and LBY-135) [113,114,115,116,117]. Several DR4-targeting antibodies have been developed. One of them is mapatumumab (Map), which has been studied in Phase II clinical trials for multiple myeloma (NCT00315757), NHL (NCT00094848), hepatocellular carcinoma (NCT01258608), cervical cancer (NCT01088347), and lung cancer (NCT00583830) [118]. Based on these studies, Map is well tolerated at 20 mg/kg/day, with a favorable safety profile [7]. However, Map has not been shown to improve response rates in cancer patients when combined with paclitaxel, gemcitabine, carboplatin, or bortezomib [7,119]. Similarly, several DR5-targeting antibodies have been developed and tested in patients with different cancers. The DR5 antibody tigatuzumab has been studied in Phase II clinical trials for NSCLC (NCT00991796), pancreatic cancer (NCT00521404), ovarian cancer (NCT00945191), colorectal cancer (NCT01124630), and triple-negative breast cancer (TNBC) (NCT01307891) [118]. Most DR5- and DR4-targeting antibodies are safe and well tolerated in patients at 20 mg/kg/day [119]. In addition, several new DR5-targeting antibodies are now entering clinical studies. The tetravalent DR5 antibody INBRX-109 is one of them, where it will be examined in a variety of solid tumors and sarcomas (NCT03715933 and NCT04950075) [120]. INBRX-109 will be studied alone and in combination with other treatment agents, including 5-fluorouracil/irinotecan- with INBRX-109 to treat pancreatic ductal cancer (NCT03715933). Another candidate is IGM-8444, which is a multimeric anti-DR5 agonist antibody that will be tested in patients with newly diagnosed, relapsed, or refractory cancers (NCT04553692) alone or in combination with other chemotherapies. Preclinical studies with IGM-8444 showed tumor cytotoxicity in vitro and in vivo with a favorable safety profile and synergized with chemotherapeutic agents, including paclitaxel and the BCL-2 inhibitor ABT-199 [121]. DR5 antibodies, such as GEN1029 and BI 905711, have also been studied in clinical trials. GEN1029 is a hexamerizing IgG that forms hexamers when it binds to its target due to a mutation of E430G in its Fc domains [122]. GEN1029 underwent Phase I and Phase II clinical studies in various malignancies (NCT03576131). However, the study’s sponsor halted the clinical trial with GEN1029 due to a narrow therapeutic window after the dose-escalation phase of the study. Lastly, BI 905711, a bispecific tetravalent antibody DR-5 and CDH17, is recruiting for Phase I clinical trials for gastrointestinal, pancreatic, and cholangiocarcinomas (NCT04137289, NCT05087992) [123]. These clinical trials were prompted by a recent preclinical study showing that BI 905711 could selectively induce apoptosis in CDH17-positive colorectal cancer cells in vitro and in vivo with a favorable safety profile [123].

Although it is unknown why DR5 antibodies outnumber DR4 antibodies, DR5 antibodies have been studied in a variety of cancers and combinations in the past [119]. Despite promising results, the inability of TRAIL death receptor antibodies to induce death receptor trimerization limits their clinical development [119,124]. Despite a favorable safety profile in patients, death receptor agonist antibodies have shown limited efficacy. Current efforts have been made to improve antibody-based therapies by modifying death receptor antibodies to enhance death receptor clustering [111]. In addition, combining with other chemotherapies is another viable strategy for improving death-receptor-based therapies. Because TRAIL and its death receptor agonist antibodies have shown limited efficacy in clinical trials, alternative strategies have been tested, including the search for small molecules that can induce TRAIL and DR5 as therapeutics. One of the promising small molecules identified using this approach is ONC201 [125]. ONC201 was initially discovered as a compound that can induce TRAIL expression and is potent against several types of tumors [126,127,128]. Subsequent studies have shown that ONC201 has several targets, including the induction of DR5 and the activation of an integrated stress response and caseinolytic protease P (CLPP) [129,130]. ONC201 has been tested in clinical trials, and its efficacy has been observed in some cancer patients, particularly those with glioblastoma that harbor H3K27M mutations [129]. ONC201 is now in Phase III clinical trials for treating adult recurrent H3 K27M-mutant high-grade glioma. 

## 8. TRAIL and Tumor Immunosurveillance

The immune system is involved in tumor prevention and elimination of tumors [131]. For example, viral infections must be eradicated to prevent virus-induced tumors. Additionally, inflammatory conditions must be resolved to prevent tumor development. In addition, tumors must be identified and removed. The TRAIL pathway has been shown to play a critical role in viral infection and tumor immune surveillance [132,133]. Increasing evidence suggests that TRAIL expression is abundant in innate and adaptive immune system cells [134]. TRAIL-expressing cells from the innate immune system include macrophages (MPs) and dendritic cells (DCs), and those from the adaptive immune system include B and T cells (CD4+ and CD8+ T cells) [135,136,137]. TRAIL is also expressed by natural killer cells (NKCs) [138]. Although TRAIL expression varies by cell types, TRAIL is stored in the intracellular pool of many immune cells from which it is secreted in response to stimuli [139,140]. Lipopolysaccharide (LPS) and pro-inflammatory cytokines, such as interferons (IFN-α, β), TNF-α, and IL2, act as stimuli by activating transcription factors that increase TRAIL transcription, resulting in increased soluble and membrane-bound TRAIL expression in immune cells [136,141,142,143,144]. 

Tumors are complex tissues composed of malignant and surrounding cells, including immune cells that interact with tumor cells [145]. TRAIL-expressing cytotoxic cells (e.g., MPs, NKCs, and T cells) and antigen-presenting cells (APCs) (e.g., DCs, MPs, and B cells) serve as the first line of defense against cancer in both the innate and adaptive immune systems [146]. Previous studies suggest that MPs recognize tumor-associated antigens (TAA) through Fcy or lectin-like receptors, as well as LDL receptor-related protein 1 (LRP1) [147,148]. Then, in response to LPS and IFN stimulation, MPs are activated, resulting in increased TRAIL expression [136,141]. Therefore, through functional interactions with immune cells, TRAIL can directly inhibit tumor cell growth by inducing cancer cell apoptosis and promoting the recruitment of immune cells (monocytes/macrophages) through chemokine secretion to kill cancer cells [146]. IFN has also been shown to increase TRAIL expression in DC [142] and NKC [132]. For DCs, IFN-stimulated TRAIL expression induces apoptosis in TRAIL-sensitive cells, including tumor cells [142]. In the case of NKCs, IFN-stimulated TRAIL induction is critical for the antitumor metastasis activity of NK cells [132]. 

## 9. TRAIL and the Immune Checkpoint PD-L1

Recent evidence suggests that the TRAIL pathway functionally interacts with the immune checkpoint PD-L1. PD-L1 is a checkpoint molecule that binds to inhibitory receptor programmed death protein 1 (PD-1) on the surface of immune cells, such as T and B cells [149]. The PD-1/PD-L1 signaling axis maintains immune homeostasis by suppressing T-cell function to prevent autoimmunity [149,150]. Many tumor cells exploit this mechanism to evade immune surveillance by overexpressing PD-L1 [151]. Although TRAIL has previously been implicated in tumor immune surveillance [133,152,153], TRAIL has been shown to induce PD-L1 expression and promote the epithelial–mesenchymal transition in the squamous cell carcinoma of the esophagus [154]. TRAIL does this by activating the ERK/STAT3 signaling pathways, which promote EMT via PD-L1 [154]. In gastric cancer cells, TRAIL can also increase the expression of PD-L1 by inhibiting miR-429 [155]. PD-L1 binds to the epidermal growth factor receptor (EGFR) and activates the cell survival signaling pathway mTOR/AKT, which reduces the sensitivity of TRAIL [155]. Furthermore, a recent study found that DR5 agonistic antibodies induce PD-L1 expression [156]. The underlying mechanism is that the activation of caspase 8 by DR5 agonistic antibodies increases Rho-associated kinase1 (ROCK1) activity to induce PD-L1 expression, which promotes immune evasion [156]. A recent study showed that PD-L1 expression confers resistance to TRAIL in tumor cells in a non-canonical manner [57]. TRAIL can induce PD-L1 expression in triple-negative breast cancer (TNBC) cells via an ERK-dependent mechanism [57]. Furthermore, PD-L1 expression has been shown to confer resistance to TRAIL in tumor cells in a non-canonical manner [57]. Inhibiting PD-L1 expression in TRAIL-resistant TNBC cells increased TRAIL sensitivity in the absence of immune cells [57]. Combining an anti-PD-L1 antibody with TRAIL effectively induced cancer cell death [157]. Thus, an increase in PD-L1 expression by the TRAIL pathway promotes EMT, cell survival, immune evasion, and TRAIL resistance (Figure 3). Based on these findings, we speculate that targeting PD-L1 may improve TRAIL/TRAIL death-receptor-based therapies in tumor cells. We also speculate that targeting the TRAIL pathway may improve PD-L1-based cancer immunotherapy.

## 10. Conclusions—Concluding Remarks

The discovery of TRAIL as a cancer-selective agent has led to extensive research into the TRAIL signaling pathway and its potential application in cancer therapy. Following promising preclinical results, TRAIL-based treatments were developed, which included recombinant TRAIL and its death receptor agonistic antibodies. While several TRAIL-based therapies have advanced to clinical trials and have been shown to be well tolerated in patients, the low efficacy of TRAIL and death receptor antibodies as cancer therapeutics limits their further development. A number of TRAIL resistance mechanisms has been identified, and treatments that target these mechanisms have been developed, including TRAIL and its death receptor antibody variants with structural modifications, nanoparticles, and novel combination strategies. Additionally, new evidence suggests that TRAIL functionally interacts with the immune checkpoint molecule PD-L1 and that targeting PD-L1 enhances the antitumor activity mediated by TRAIL. These findings are encouraging because they capitalize on TRAIL’s role in tumor immunosurveillance and pave the way for clinical trials by combining TRAIL-based therapies with immune checkpoint blockers. Finally, a better understanding of the pathways that regulate the TRAIL pathway in the context of the immune checkpoint PD1/PD-L1 axis may lead to improving TRAIL-based therapy. Thus, it is conceivable that understanding these issues will aid in better developing TRAIL-based cancer therapy.

## Figures and Tables

**Figure 1 cancers-15-02752-f001:**
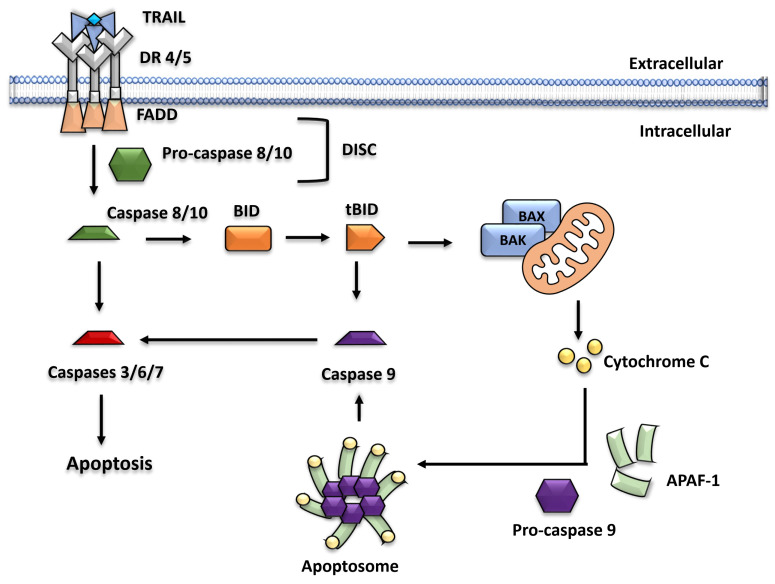
Schematic overview of the TRAIL apoptosis pathway. TRAIL induces apoptosis by binding to death receptors 4/5. This promotes the trimerization of the death receptor, which recruits FADD and pro-caspase 8/10 to form DISC. The latter activates caspase 8/10, which activates caspase 3 to induce apoptosis. Caspase 8/10 can also convert BID to tBID by cleavage, increasing apoptosis. To accomplish this, tBID translocates to the mitochondria, activating BAX/BAK. The formation of mitochondrial pores is then mediated by activated BAX/BAK, resulting in the release of cytochrome c. An apoptosome is formed when Apaf-1 and pro-caspase 9 bind together. Caspase 9 is activated, which then activates caspases 3, 6, and 7 to increase apoptosis.

**Figure 2 cancers-15-02752-f002:**
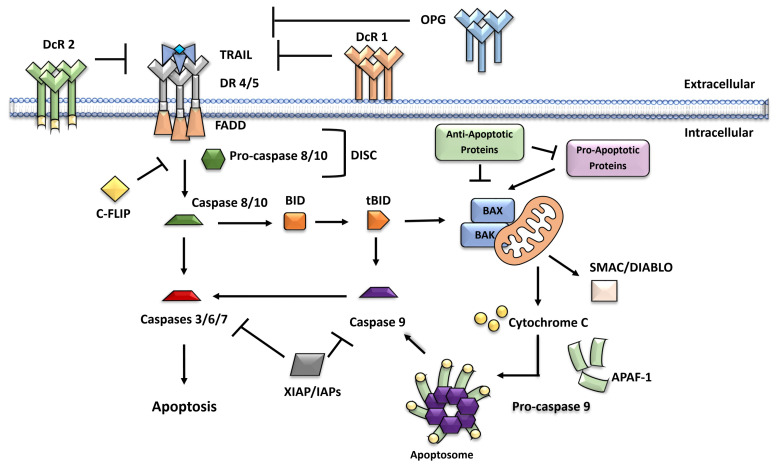
Mechanisms of TRAIL resistance. TRAIL resistance mechanisms can occur at any point along the TRAIL signaling pathway. Overexpression of decoy receptors, such as DcR1, DcR2, and OPG, can inhibit TRAIL-induced activation of DR4 and DR5 at the cell surface. TRAIL-induced apoptosis can also be inhibited by the inhibitory protein c-FLIP. This is accomplished by c-FLIP binding to FADD or procaspases 8/10 and inhibiting DISC formation. Antiapoptotic proteins (e.g., BCL-2, BCL-XL) can inhibit the TRAIL signaling pathway. These proteins bind to pro-apoptotic proteins (e.g., BAX), preventing MOMP, SMAC/DIABLO, and cytochrome c release. Two more downstream TRAIL signaling inhibitors are XIAP and IAP. These proteins prevent caspase 9, as well as caspases 3, 6 and 7 activation, which in turn inhibits apoptosis.

**Figure 3 cancers-15-02752-f003:**
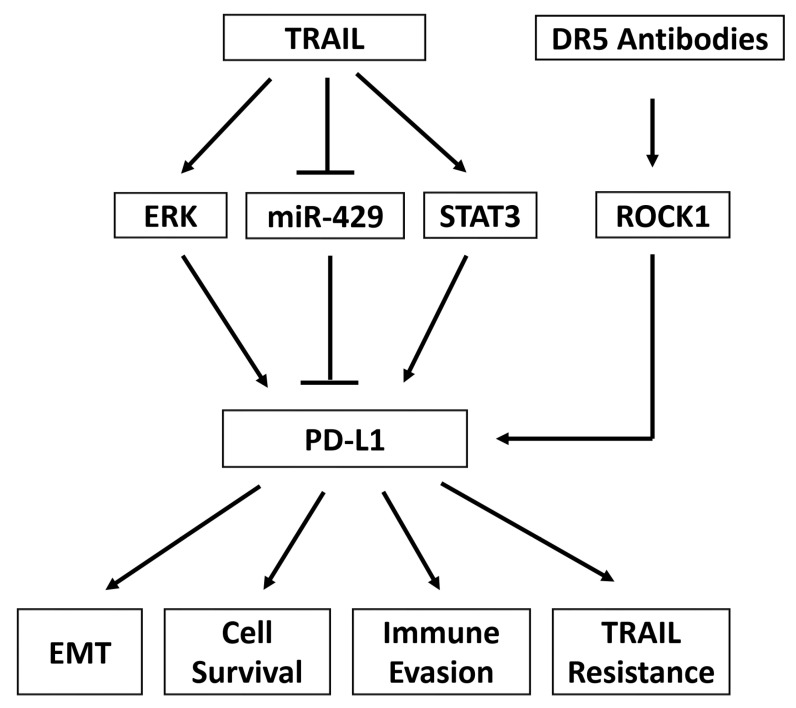
TRAIL and DR5 agonist antibodies increase PD-L1 expression. TRAIL can induce PD-L1 expression by activating the ERK and STAT3 pathways or inhibiting miR-429. DR5 agonist antibodies can induce PD-L1 expression by activating ROCK1. Induction of PD-L1 then results in EMT, cell survival, immune evasion, and resistance to TRAIL.

**Table 1 cancers-15-02752-t001:** Clinical trials of rTRAIL and death-receptor-targeting antibodies.

Type	Name	Cancer	Phase	Clinical Trial ID
rTRAIL	Dulanermin	Non-small cell lung cancer	III	NCT03083743
			II	NCT00508625
		Non-Hodgkin’s lymphoma	II	NCT00400764
		Colorectal cancer	I	NCT00671372
	SCB-313	Peritoneal malignancies	I	NCT03443674
		Peritoneal carcinomatosis	I	NCT04047771
	ABBV-621	Advanced solid tumors and hematological malignancies	I	NCT03082209
		Multiple myeloma	II	NCT04570631
DR4 targeting	Mapatumumab	Multiple myeloma	II	NCT00315757
		Non-Hodgkin’s lymphoma	II	NCT00094848
		Hepatocellular carcinoma	II	NCT01258608
		Non-small cell lung cancer	II	NCT00583830
		Cervical cancer	II	NCT01088347
DR5 targeting	Tigatuzumab	Non-small cell lung cancer	II	NCT00991796
		Pancreatic cancer	II	NCT00521404
		Triple-negative breast cancer	II	NCT01307891
		Ovarian cancer	II	NCT00945191
		Colorectal cancer	I	NCT01124630
	INBRX-109	Solid tumors, malignant pleura mesothelioma, gastric, colorectal, sarcoma (Ewing and chondrosarcoma), pancreatic	I	NCT03715933
		Chondrosarcoma	II	NCT04950075
	IGM-8444	Solid tumors, colorectal, lymphoma (non-Hodgkin and small lymphocytic), sarcoma (chondrosarcoma), leukemia (chronic lymphocytic and acute)	I	NCT04553692
	GEN1029	Colorectal, non-small cell lung, triple-negative breast, renal cell, gastric, pancreatic, and urothelial cancers	II	NCT03576131
	BI 905711	Gastrointestinal, pancreatic, and cholangiocarcinoma	I	NCT04137289
		Gastrointestinal cancers	I	NCT05087992

## Data Availability

All representative data are contained within the article.

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
