# Peer review of "The Role of TRAIL in Apoptosis and Immunosurveillance in Cancer"

_cancers, 2023, doi:10.3390/cancers15102752_

Round 1

Reviewer 1 Report

Thank you for giving me the opportunity to review the manuscript “The Role of TRAIL in Apoptosis and Immunosurveillance in Cancer” intended for publication in Cancers, (section: tumor microenvironment, special issue: unique perspectives in cancer signaling).

The review of Pimentel et al. is divided into the following sections: a general overview over the ligand TRAIL and its receptors followed by a description of the TRAIL apoptosis pathway and resistance mechanisms. Furthermore, the authors discuss the role of TRAIL-based therapy in cancer and its involvement in tumor immunosurveillance. Last but not least, the authors give an outlook and briefly explain the interaction of TRAIL and the immune checkpoint molecule PD-L1.

In my opinion, there are some limitations:

The manuscript mainly consists of a repetition of general issues concerning TRAIL and cancer therapy that have been frequently covered in previous reviews in the last ten years (even in the journal Cancers).

It must also be noted here that Pimentel et al. miss the important point that non-apoptotic TRAIL signaling pathways are particularly relevant with respect to cancer therapy and resistance mechanisms. In TRAIL apoptosis resistant tumors, non-apoptotic signaling (e.g. NFkB signaling pathway) can stay intact, leading to pro-tumoral activity of TRAIL. Indeed, this has been reported in several preclinic tumor models (e.g. Hartwig et al. 2017, DOI: 10.1016/j.molcel.2017.01.021; von Karstedt et al. 2015, DOI: 10.1016/j.ccell.2015.02.014).

Interesting and novel for a TRAIL review is the relationship of TRAIL and PDL1. Unfortunately, I miss in this manuscript the underlying background and perspectives for therapies targeting these findings, e.g. combining TRAIL-based therapies with immune checkpoint blockers. The section “TRAIL and the immune checkpoint PD-L1” only touches on the topic and takes only up half a page in the review. Furthermore, the authors claim that PD-L1 expression has been shown to confer resistance to TRAIL in tumor cells in a non-canonical manner (Pimentel et al., 2023). Another reason why I don't understand that the important aspect of non-apoptotic TRAIL signaling is not considered in more detail in the review.

Further, the authors mainly refer to publications that were published before 2020, the majority of them even before 2010. As of 2020, only 10 publications are cited and 7 of these are reviews. Thus, too few current original articles are cited, summarized and covered by this review.

Author Response

  • The manuscript mainly consists of a repetition of general issues concerning TRAIL and cancer therapy that has been frequently covered in previous reviews in the last ten years (even in the journal Cancers).

Response: Although TRAIL and cancer therapy have been discussed repeatedly in several reviews, these issues are fundamental and necessary for the background information that leads to the topic discussed in this manuscript.

  • Pimentel et al. miss the important point that non-apoptotic TRAIL signaling pathways are particularly relevant with respect to cancer therapy and resistance mechanisms. In TRAIL apoptosis resistant tumors, non-apoptotic signaling (e.g. NFkB signaling pathway) can stay intact, leading to pro-tumoral activity of TRAIL. Indeed, this has been reported in several preclinic tumor models (e.g. Hartwig et al. 2017, DOI: 10.1016/j.molcel.2017.01.021; von Karstedt et al. 2015, DOI: 10.1016/j.ccell.2015.02.014).

Response: TRAIL resistance is heterogenous and involves a number of pathways, including the activation of survival pathways, such as the NF-kB pathway. While we briefly touched on some of these issues without detailed specificity. Per this reviewer’s suggestion, we have added a new section describing the non-apoptotic signaling pathway activated by the TRAIL pathway.

  • The reviewer stated that the section “TRAIL and the immune checkpoint PD-L1” only touches on the topic and takes only up half a page in the review. Furthermore, the authors claim that PD-L1 expression has been shown to confer resistance to TRAIL in tumor cells in a non-canonical manner (Pimentel et al., 2023). In addition, this reviewer stated that the important aspect of non-apoptotic TRAIL signaling is not considered in more detail in the review.

Response: Although several previous publications have linked TRAIL/agonist antibodies against TRAIL death receptors to PD-L1 expression, the role of TRAIL/death receptors in PD-L1 expression involves the immune system. Our recent study suggests that PD-L1 may play a role in TRAIL resistance via a non-canonical pathway (Pimentel et al., 2023), but the underlying mechanisms are unknown. The non-canonical role of PD-L1 in TRAIL resistance is a fairly new concept that needs further investigation. Thus, we have added a more detailed discussion regarding the regulation of PD-L1 by TRAIL and DR5 antibodies (New Fig. 3). As for non-apoptotic TRAIL signaling, we have addressed it in #1 (above).

4). Further, the authors mainly refer to publications that were published before 2020, the majority of them even before 2010. As of 2020, only 10 publications are cited and 7 of these are reviews. Thus, too few current original articles are cited, summarized and covered in this review.

Response: We agree that many of the studies cited in this article are from before 2020. Many original studies on the TRAIL apoptosis pathway and TRAIL resistance were discovered before 2010. While several recent publications described some new aspects of TRAIL biology, most publications have been translational since 2010. Nonetheless, we have added several recently original publications in this revision.

Reviewer 2 Report

The review manuscript “The role of TRAIL in apoptosis and immunosurveillance in cancer” submitted by Pimentel et al. aims to overview both TRAIL-induced apoptotic signaling mainly in cancer cells and the status of anticancer therapeutic approaches involving TRAIL-related compounds. In general, the information on both TRAIL-induced signaling and the past and current status of TRAIL-based cancer therapy trials and possible novel TRAIL-related agents is presented in the manuscript fairly superficially and in several instances leading to confusion and even incorrect statements. An example of a confusing statement in chapter 3 is – “DR4 and DR5 are type 1 transmembrane proteins with an intracellular death domain (DD) connecting 2-4 cys-rich CTE domains”. Cysteine-rich domains form the extracellular of DR4 and DR5 and thus not directly linked to the intracellular DD. CTE according to the list of abbreviations meaning C-terminal extracellular reflects the extracellular part of TRAIL ligand as a type II transmembrane protein and not of DR4 or DR5 receptors… In the part of the manuscript dealing with exploring TRAIL-induced apoptosis in cancer therapy are not even mentioned novel approaches exploring TRAIL-containing nanoparticles or liposomes – see current review in Cancers 2022, 14, 5125. https://doi.org/10.3390/cancers14205125 . Thus in my opinion this manuscript is not suitable for publication in the Cancers journal.

Author Response

  • A concern regarding the statement of “An example of a confusing statement in chapter 3 is – “DR4 and DR5 are type 1 transmembrane proteins with an intracellular death domain (DD) connecting 2-4 cys-rich CTE domains”. Cysteine-rich domains form the extracellular of DR4 and DR5 and thus not directly linked to the intracellular DD. CTE according to the list of abbreviations meaning C-terminal extracellular reflects the extracellular part of TRAIL ligand as a type II transmembrane protein and not of DR4 or DR5 receptors.”

Response: We agree that this statement is incorrect and has been corrected as the following sentence: DR4 and DR5 are type 1 transmembrane proteins consisting of a cysteine-rich extracellular domain, a transmembrane domain, and an intracellular domain with a death domain. In addition, we have deleted the Abbreviation “CTE” from the section describing death receptors to reflect type I receptors.

  • This reviewer suggested including nanoparticle or liposome approaches to deliver TRAIL into cancer cells.

Response: We have included these two approaches in the text.

Reviewer 3 Report

Dear Authors,

Please find below my main concerns and comments that I hope will help you produced a publishable revised manuscript. 

#1 : Chapter 4 a bit too long with respect to the intrinsic pathway. This chapter could be updated with considerations related to caspase-8 activation, such as for instance how TRAIL or agonist antibodies need to be prepared or formulated to be effective. This will make sense and help you discussing the ongoing clinical trials, see below. You could also comment and include princeps papers describing or demonstrating that not all cancer cells transduce TRAIL-induced apoptosis similarly through DR4 and DR5.

#2 Along the line, princeps papers need to be cited. There are too many concepts or important sentences in your review pointing to reviews. Please cite original papers. See for example chapter 7, you write “… The TRAIL pathway has been shown to play a critical role in viral infection and tumor immune surveillance (100,101). …” Yet, ref 100, 101 are review papers. There should be here a citation to the original papers (116 and 120). These important concepts are not sufficiently and appropriately presented in this chapter. 

Similar comment applies to the review for which references often omit citing original papers, citing instead review papers. This need to be carefully checked and corrected.

#3 The chapter describing clinical trials is not comprehensive enough and for sure not up-to-date ! A quick look at the ongoing clinical trials highlight the poor analysis reported in your review : see some examples below, which could be discussed and put into perspective with  comment #1. See below

Clinical trials also include cancer or non cancer assessement of TRAIL see below:

NCT05571371 : Inflammatory Biomarkers in Psychogenic Non-epileptic Seizure 

Authors can also have a look at the following ongoing clinical trials using multimeric moabs for cancer therapy

NCT03715933 INBRX-109 a recombinant humanized tetravalent antibody targeting the human death receptor 5 (DR5).

Or NCT04553692 another anti_DR5 moAb formulated as an IgM

Not mentioning the Abbvie ABBV-621 that the Authors ought to discuss should they like to provide a minimal overview of ongoing and past clinical trials. 

You should in my opinion explain your choice to produce Table 1 with so few of the PARAs (pro-apoptotic Receptor activators). 

#4 : Chapters 7 and 8 should be more elaborated and comprehensive for the readers. You should in my opinion include a figure summarizing the findings describing the link that you’d like to put forward in your review, that is to say the relationship or molecular circuitry linking TRAIL and PD-L1 .

Author Response

  • Chapter 4 a bit too long with respect to the intrinsic pathway. This chapter could be updated with considerations related to caspase-8 activation, such as for instance how TRAIL or agonist antibodies need to be prepared or formulated to be effective. You could also comment and include princeps papers describing or demonstrating that not all cancer cells transduce TRAIL-induced apoptosis similarly through DR4 and DR5.

Response:  We agree with this reviewer Chapter 4 is too long. We have shortened it. Concerning the preparation or formulation of TRAIL or agonist antibodies, we have discussed it under the chapter on Targeting TRAIL and TRAIL death receptors for cancer therapies. As for TRAIL signaling in cancer cells, preferentially DR4 or DR5, we have added a more detailed discussion regarding using DR4 and DR5 in TRAIL-induced apoptosis.

  • This reviewer suggested citing original papers not review articles, including the following statement “The TRAIL pathway has been shown to play a critical role in viral infection and tumor immune surveillance (100,101). …” Yet, ref 100, 101 are review papers. There should be here a citation to the original papers (116 and 120).

Response: We agree that original research papers should be cited in the text and have cited those two original research papers. While we intended to cite original research papers, some of the review articles we cited cover broad topics.

  • This reviewer raised a concern regarding not including updated clinical trials including NCT05571371, NCT03715933, NCT04553692 and Abbvie ABBV-621.

Response: We have included these clinical trials excerpts for NCT05571371(non-cancer trial) because we focus this review on cancer. We have updated the table by including a complete list of clinical trials.

  • This reviewer suggested including a diagram describing the relationship or molecular circuitry linking TRAIL and PD-L1.

Response: We appreciate the reviewers' enthusiasm in suggesting a figure demonstrating the relationship or molecular circuitry linking TRAIL and PD-L1. Therefore, we have included Fig. 3 (new figure).

Round 2

Reviewer 1 Report

Thank you for the given update of the manuscript “The Role of TRAIL in Apoptosis and Immunosurveillance in Cancer” intended for publication in Cancers, (section: tumor microenvironment, special issue: unique perspectives in cancer signaling).

Although I recommended rejecting the manuscript in the first review round, the authors were given the chance to make profound changes to the paper. In their new version, Cragg et al have endeavored to address the criticisms I noted. Nevertheless, in my opinion, further revision of the manuscript is needed. In the following I would like to make my comments on the current version and answer to the authors comments.

1)        The manuscript mainly consists of a repetition of general issues concerning TRAIL and cancer therapy that has been frequently covered in previous reviews in the last ten years (even in the journal Cancers).

Response: Although TRAIL and cancer therapy have been discussed repeatedly in several reviews, these issues are fundamental and necessary for the background information that leads to the topic discussed in this manuscript.

I continue to believe that the review by Pimentel et al. repeats mainly general issues concerning the role of TRAIL in cancer therapy. But I agree with the authors that in order to understand the background, certain fundamentals have to be presented. Fortunately, the authors have made some cuts in appropriate places so as not to lengthen the manuscript unnecessarily.

2)        Pimentel et al. miss the important point that non-apoptotic TRAIL signaling pathways are particularly relevant with respect to cancer therapy and resistance mechanisms. In TRAIL apoptosis resistant tumors, non-apoptotic signaling (e.g. NFkB signaling pathway) can stay intact, leading to pro-tumoral activity of TRAIL. Indeed, this has been reported in several preclinic tumor models (e.g. Hartwig et al. 2017, DOI: 10.1016/j.molcel.2017.01.021; von Karstedt et al. 2015, DOI: 10.1016/j.ccell.2015.02.014).

Response: TRAIL resistance is heterogenous and involves a number of pathways, including the activation of survival pathways, such as the NF-kB pathway. While we briefly touched on some of these issues without detailed specificity. Per this reviewer’s suggestion, we have added a new section describing the non-apoptotic signaling pathway activated by the TRAIL pathway.

In my opinion, it was important to include the TRAIL-mediated non-apoptotic signaling pathways in more detail in the manuscript. Pimentel et al. have implemented that satisfactory.

1)     The reviewer stated that the section “TRAIL and the immune checkpoint PD-L1” only touches on the topic and takes only up half a page in the review. Furthermore, the authors claim that PD-L1 expression has been shown to confer resistance to TRAIL in tumor cells in a non-canonical manner (Pimentel et al., 2023). In addition, this reviewer stated that the important aspect of non-apoptotic TRAIL signaling is not considered in more detail in the review.

Response: Although several previous publications have linked TRAIL/agonist antibodies against TRAIL death receptors to PD-L1 expression, the role of TRAIL/death receptors in PD-L1 expression involves the immune system. Our recent study suggests that PD-L1 may play a role in TRAIL resistance via a non-canonical pathway (Pimentel et al., 2023), but the underlying mechanisms are unknown. The non-canonical role of PD-L1 in TRAIL resistance is a fairly new concept that needs further investigation. Thus, we have added a more detailed discussion regarding the regulation of PD-L1 by TRAIL and DR5 antibodies (New Fig. 3). As for non-apoptotic TRAIL signaling, we have addressed it in #1 (above).

The relationship of TRAIL and PDL1 is very interesting and the authors discuss the underlying background in more detail in the new version of the manuscript. It would still be desirable for the authors to give an outlook or speculations on combining TRAIL-based therapies with immune checkpoint blockers. In Figure 3 it should be added that DR5 antibodies are agonistic.

4). Further, the authors mainly refer to publications that were published before 2020, the majority of them even before 2010. As of 2020, only 10 publications are cited and 7 of these are reviews. Thus, too few current original articles are cited, summarized and covered in this review.

Response: We agree that many of the studies cited in this article are from before 2020. Many original studies on the TRAIL apoptosis pathway and TRAIL resistance were discovered before 2010. While several recent publications described some new aspects of TRAIL biology, most publications have been translational since 2010. Nonetheless, we have added several recently original publications in this revision.

I have noted that the authors have incorporated more recent references. This also enhances Table 1 (clinical trials) and makes the manuscript more up-to-date.

Further major points:

-     1. There is a lot of repetition in the simple summary and abstract. it makes sense to make the simple summary even more compact and shorter.

-       2. Figure 2: the drawing is not clear. As it is now, the impression is given that the decoy receptors directly inhibit the DR4/5 receptors. The important point is that the decoy receptors intercept TRAIL and thus there is no more or less TRAIL available to bind to DR4/5.  

Other minor but important points:

-        - uniform font size in section 2. TRAIL ligand

-        - uniform spelling type xy proteins: 1 or I / 2 or II

-        - the abbreviation (DD) is missing at the beginning of section 3. TRAIL receptors

-        - Format of abbrevation table has to be updated

Author Response

  • This reviewer believes that this MS repeats mainly general issues concerning the role of TRAIL in cancer therapy, although it is acknowledged that the authors have made some cuts in appropriate places so as not to lengthen the manuscript unnecessarily.

Response: We also did an additional cut to keep the manuscript from becoming too long.

  • The reviewer suggested including the TRAIL-mediated non-apoptotic signaling pathways in more detail in the manuscript.

Response: We agreed with the reviewer that TRAIL-mediated non-apoptotic pathways are important aspects of TRAIL biology and need to be discussed in more detail. Because this MS focuses on the role of TRAIL in apoptosis and immune surveillance, we tried to balance the length of the content of apoptosis/immune surveillance vs. nonapoptotic signaling pathways.

  • The reviewer suggested that an outlook or speculations be made about combining TRAIL-based therapies with immune checkpoint blockers. In Figure 3 it should be added that DR5 antibodies are agonistic.

Response: We have added two sentences to speculate that combining TRAIL-based therapies with immune checkpoint blockers may improve cancer therapies. We've also corrected Figure 3 to show that DR5 antibodies are agonistic.

  • This reviewer noted repetition in the simple summary and abstract.

Response: We have revised the simple summary to make it more concise and reduce repetition compared to the abstract.

  • Figure 2: The drawing is not clear. As it is now, the impression is given that the decoy receptors directly inhibit the DR4/5 receptors. The important point is that the decoy receptors intercept TRAIL, and thus there is no more or less TRAIL available to bind to DR4/5.  

Response: We have revised Figure 2 to show that decoy receptors impede TRAIL-DR binding by intercepting TRAIL.

  • Uniform font size in Section TRAIL ligand

Response: We have made changes.

  • Uniform spelling type xy proteins: 1 or I / 2 or II

Response: We have made such changes as the reviewer suggested.

  • The abbreviation (DD) is missing at the beginning of Section TRAIL receptors

Response: We have made this change.

  • The format of the abbreviation table has to be updated.

Response: We have updated the format of the abbreviation table.

Reviewer 2 Report

Surprisingly, the authors greatly improved the previous unacceptable version of the manuscript, and in its current corrected and improved version it is acceptable for publication. 

Author Response

Thank you for accepting our paper.

Reviewer 3 Report

Thanks for addressing my comments. The review is now acceptable for publication.

Author Response

Thank you for accepting our paper.

Round 3

Reviewer 1 Report

I have no further comments and accept the current version of the manuscript.